# Use of oral polio vaccine and the incidence of COVID-19 in the world

**Farrokh Habibzadeh**[1,2], **Konstantin Chumakov**[3], **Mohammad M. Sajadi**[4,5],
**Mahboobeh Yadollahie**[6], **Kristen Stafford**[4,5], **Ashraf Simi**[2], **Shyamasundaran Kottilil**[4,5],
**Iman Hafizi-Rastani**[2], **Robert C. Gallo**[4,5]*

**1** Global Virus Network, Middle East Region, Shiraz, Iran, **2** Research and Development Headquarters, Petroleum Industry Health Organization, Shiraz, Iran, **3** Office of Vaccines Research and Review, Food and Drug Administration, Global Virus Network Center of Excellence, Silver Spring, Maryland, United States of America, **4** Institute of Human Virology, University of Maryland School of Medicine, Baltimore, Maryland, United States of America, **5** Global Virus Network, Baltimore, Maryland, United States of America, **6** Freelance Researcher, Shiraz, Iran

* rgallo@ihv.umaryland.edu

## Abstract

### Background

Several live attenuated vaccines were shown to provide temporary protection against a variety of infectious diseases through stimulation of the host innate immune system.

### Objective

To test the hypothesis that countries using oral polio vaccine (OPV) have a lower cumulative number of cases diagnosed with COVID-19 per 100,000 population (CP100K) compared with those using only inactivated polio vaccine (IPV).

### Methods

In an ecological study, the CP100K was compared between countries using OPV *vs* IPV. We used a random-effect meta-analysis technique to estimate the pooled mean for CP100K. We also used negative binomial regression with CP100K as the dependent variable and the human development index (HDI) and the type of vaccine used as independent variables.

### Results

The pooled estimated mean CP100K was 4970 (95% CI 4030 to 5900) cases per 100,000 population for countries using IPV, significantly (p<0.001) higher than that for countries using OPV—1580 (1190 to 1960). Countries with higher HDI prefer to use IPV; those with lower HDI commonly use OPV. Both HDI and the type of vaccine were independent predictors of CP100K. Use of OPV compared to IPV could independently decrease the CP100K by an average of 30% at the mean HDI of 0.72.

**Data Availability Statement:** The dataset is available as supplementary material to this article.

**Funding:** The authors received no specific funding for this work.

**Competing interests:** The authors have declared
that no competing interests exist.

## Conclusions

Countries using OPV have a lower incidence of COVID-19 compared to those using IPV. This might suggest that OPV may either prevent SARS-CoV-2 infection at individual level or slow down the transmission at the community level.

## Introduction

Coronavirus disease 2019 (COVID-19) has become a major global health concern. By February 13, 2022, it has affected more than 410 million people worldwide; the death toll exceeds 5.8 million [1]. Many countries have begun mass vaccination of their people with a handful of specific vaccines developed thus far that have received authorization. However, there are limitations in the production and distribution of the SARS-CoV-2 specific vaccines worldwide, particularly in resource-limited countries. Furthermore, the specific vaccines currently available were made using the original strain of the severe acute respiratory syndrome coronavirus 2 (SARS-CoV-2), which may not be effective against the emerging virus variants [2].

Several studies have shown that the host innate immune system has an important role in combating SARS-CoV-2 [3,4]. "Live" attenuated vaccines (LAVs) stimulate the innate immune system which may provide temporary protection against other viruses [5–9]. A systematic review of 10 cohort and two case-control studies revealed that administration of measles vaccine to children residing in seven developing countries decreased the all-cause mortality by 30% to 86%, far more than that was anticipated based on protection produced solely against measles [8]. Attenuated bacterial vaccines such as Bacillus Calmette-Guérin (BCG) can also provide non-specific immunity against unrelated diseases [10–12]. A recent study reveals that BCG vaccination can not only protect newborns against tuberculosis, but it also provides protection against non-tuberculous infectious disease during the neonatal period [13]. Meta-analysis of five clinical trials showed that BCG vaccination was associated with 30% reduction in all-cause mortality in children under 5 years of age [9]. A recent study revealed that stimulation of the innate immune system by human rhinovirus could block SARS-CoV-2 virus replication through triggering interferon production [14]. It was shown that administration of oral polio vaccine (OPV) to more than 60,000 people resulted in an almost 4-fold decrease in the morbidity and mortality associated with influenza; no clinically important side effects were reported [15,16].

OPV is an LAV. We hypothesized that receiving OPV can prevent COVID-19. Conducting a quality randomized clinical trial to test this hypothesis is not easy, partly because OPV supply is mainly used for the high-priority WHO global polio eradication initiative. However, it is possible to indirectly test the hypothesis at a population level. OPV is made from live attenuated virus that can transmit to contacts of vaccine recipients. In a recent cohort study, we have shown that indirect exposure to the attenuated poliovirus excreted by OPV recipients was associated with decreased symptomatic infection with COVID-19 [17]. Herein, we would like to test the hypothesis in an alternative indirect way. We hypothesized that countries using OPV have a lower cumulative number of cases diagnosed with COVID-19 per 100,000 population (CP100K) compared with those using inactivated polio vaccine (IPV) only.

We thus conducted this study to compare the CP100K between countries using OPV and IPV. There are many other variables that might affect the incidence of the disease, including the quality of the health care system and the surveillance infrastructures that would certainly

influence the detection rate and reporting of cases with COVID-19. Therefore, an important part of this study was to account for confounding factors that might influence the conclusions.

## Materials and methods

This ecological study was based on the data about the cumulative number of COVID-19 cases diagnosed in each country, the country population and population density, the median age and the life expectancy at birth, the gross domestic product (GDP) per capita, human development index (HDI), and the type of polio vaccine used in each country. We also retrieved the stringency index, a composite metric calculated from 9 response indicators (*e.g.*, school closures, workplace closures, travel bans), ranging from 0 (no restriction) to 100 (highest levels of restrictions). The index was retrieved for each country for each day before April 9, 2021, when all the data were retrieved from "Our World in Data" website [18,19]. The mean stringency index for each country was used for data analyses. Data about the type of polio vaccine used by each country were provided by the World Health Organization Global Polio Eradication Initiative (GPEI). The data for generating the world map were retrieved from Natural Earth, a public domain map dataset [20].

### Ethics

The study protocol was approved by the Petroleum Industry Health Organization (PIHO) R&D Institutional Review Board.

### Statistical analysis

*R* software version 4.1.0 (2021-05-18) was used for data analysis. Wilcoxon rank sum test was used to compare the distribution of two continuous variables not normally distributed. Continuous variables were expressed as median (interquartile range [IQR]) or mean (95% confidence interval [CI]). Spearman's $\rho$ was used to determine the extent of correlation between the continuous variables.

The CP100K was calculated by dividing the number of cases diagnosed in each country by its population at the midpoint of the study period multiplied by 100,000. Pooled estimation of CP100K for countries using IPV and OPV was calculated by a random-effect meta-analysis model, employing the *R* package *metafor*. Because CP100K (dependent variable in our analysis) had overdispersion, negative binomial regression analysis was used (with function *glm.nb* of the *R* package *MASS*). The median age, life expectancy at birth, and GDP per capita had a high correlation with HDI; thus, we only considered HDI and the type of polio vaccine used in each country as independent variables in the model. Interaction of the two independent variables was also examined. A p value <0.05 was considered statistically significant.

## Results

Data on the type of polio vaccine used were available for 202 countries—56 used IPV only and 146 used OPV in combination with IPV. The CP100K reported from countries using IPV only was higher than that in countries using OPV (Figs 1 and 2). The pooled estimated mean CP100K, derived using a random-effect ($I^2 = 1.0$) meta-analysis model, for countries using IPV was 4970 (95% CI 4030 to 5900) cases per 100,000 population, significantly (p<0.001) higher than that for countries using OPV—1580 (1190 to 1960).

The distributions of the median age, life expectancy, and GDP per capita were significantly different between countries using IPV and OPV (Table 1). The median HDI in countries using IPV was significantly (0.90 *vs* 0.70, p<0.001) higher than that in countries using OPV. The

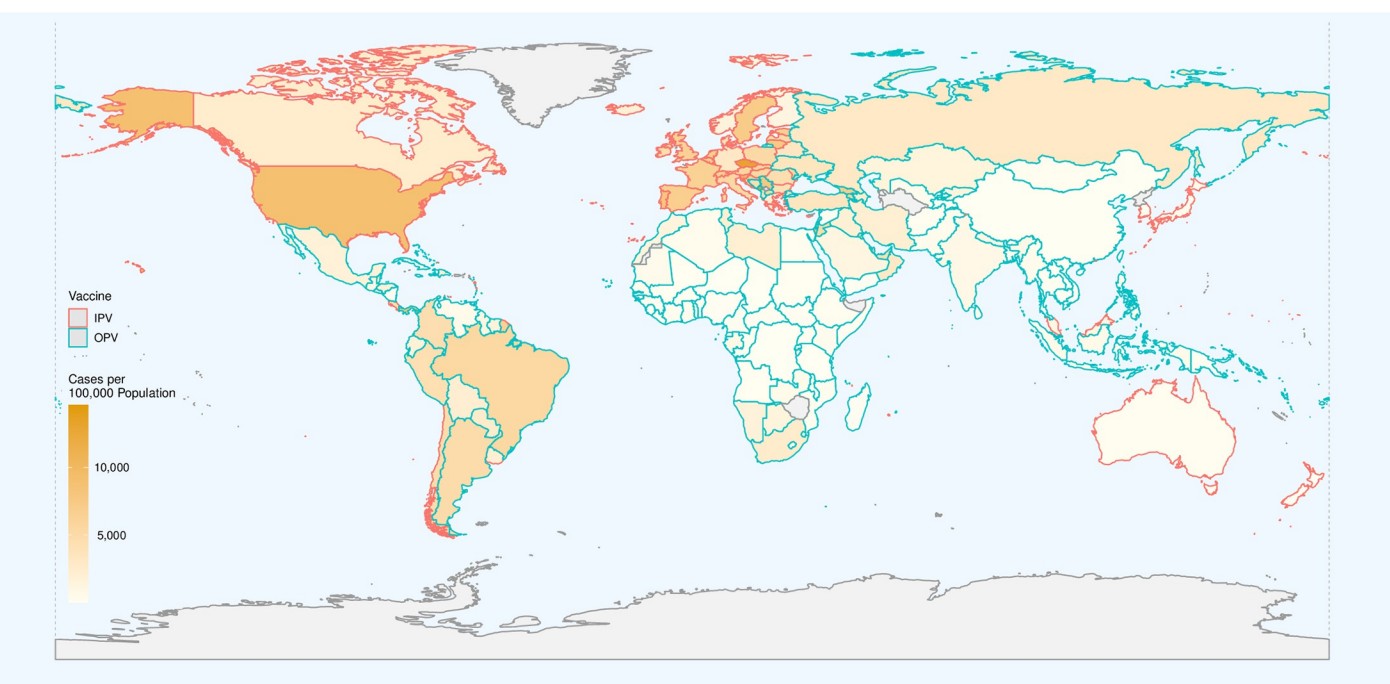

**Fig 1. The distribution of COVID-19 incidence rate in each country.** The fill color reflects the cumulative number of cases diagnosed with the disease per 100,000 population. The countries' border color shows the type of polio vaccine used in each country. Complete data were not available for gray areas. Made with Natural Earth (https://www.naturalearthdata.com/).

population density and the mean stringency index were not significantly different between countries using OPV and those using IPV. The median age, life expectancy at birth, and the GDP per capita, all had a significant (p<0.001) high correlation ($\rho$>0.92) with HDI, the composite index that accounts for these and some other parameters (Fig 3). To avoid multicollinearity, we have only used HDI and the type of vaccine as independent variables in the regression model. CP100K, the dependent varaible in our model, had a mean of 2496 cases per 100,000; the variance exceeded $9\times10^6$. For overdispersion, we used a negative binomial regression.

The model could explain more than 70% (Nagelkerke's $R^2$ = 0.734) of the variance observed in CP100K (Table 2). It showed an interaction between HDI and the type of vaccine used. Use of OPV compared to IPV could independently decrease the CP100K by an average of 30% at the mean HDI of 0.72 (Table 2); the protection provided was higher for countries with lower HDI. For example, the value corresponding to an HDI of 0.55, the 25[th] percentile HDI in countries using OPV (Table 1), was 75%. HDI was also an independent predictor of CP100K; each 0.1 unit increase in HDI increased CP100K by an average of 2.47-fold; it was 1.62-fold for countries using IPV and 2.90-fold for countries using OPV (Table 2).

## Discussion

Use of OPV by a country was an independent predictor for a lower CP100K diagnosed and reported in that country. The pooled mean CP100K was significantly lower in countries using OPV compared with those using IPV. However, this was the result of a univariate analysis and could be affected by other variables. For example, the observed difference might be attributed to the limited availability of diagnostic tests (*e.g.*, RT-PCR) in the low- and middle-income

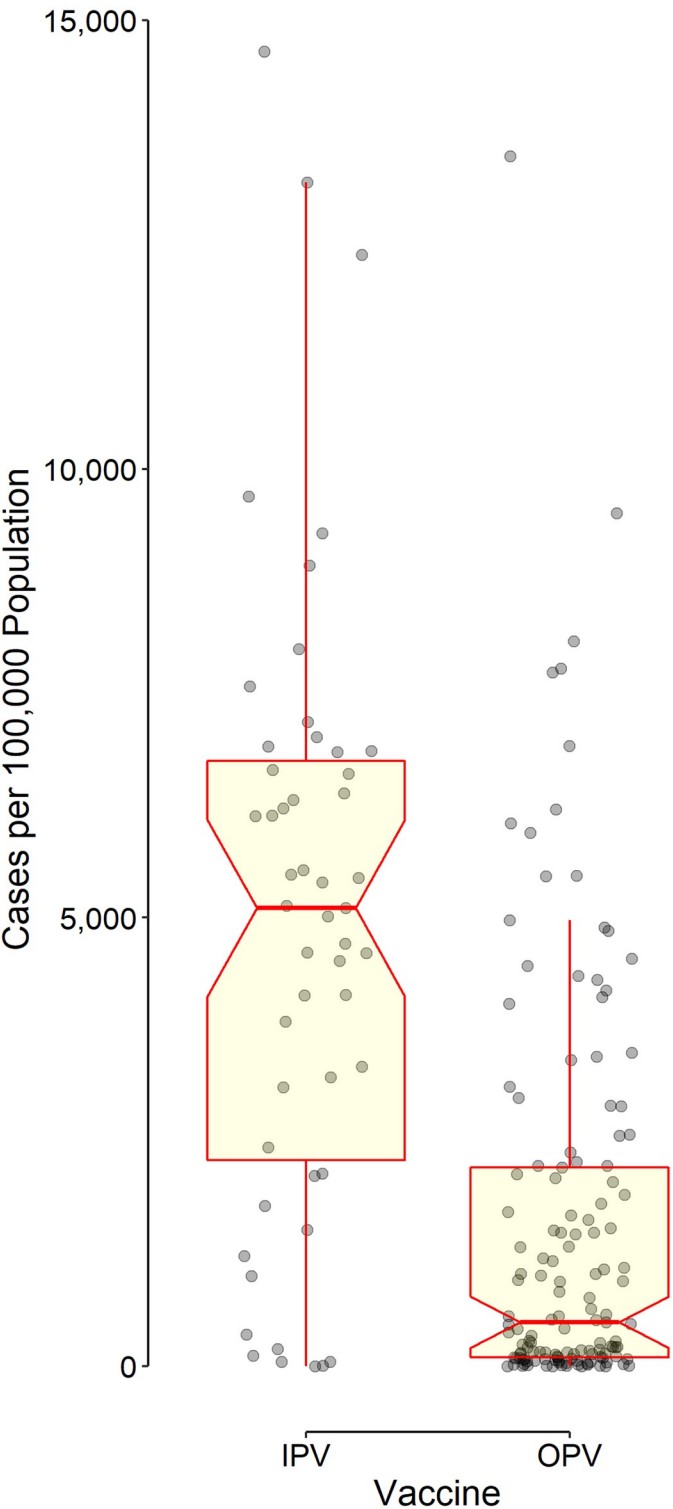

**Fig 2. Distribution of data points as well as the box and whisker plot indicating the cumulative number of cases diagnosed with COVID-19 per 100,000 population stratified by the type of polio vaccine used in each country.** The horizontal line in the middle of each box indicates the median. The notch represents the 95% confidence interval of the median. The bottom and top borders of the box show the 25th and 75th percentiles, respectively. The lower whisker indicates the smallest data point within 1.5 times the interquartile range (IQR) less than the 25th percentile; the upper whisker indicates the largest point within 1.5 × IQR greater than the 75th percentile. Points greater than the upper whisker and smaller than the lower whisker were considered outliers. All outliers were included in data analyses.

**Table 1. Median (IQR) of studied continuous variables stratified by the type of polio vaccine used.**

| Variable | Type of polio vaccine used | | p value |
|---|---|---|---|
| | IPV (n = 56) | OPV (n = 146) | |
| Median age, yrs | 42 (38, 44) | 26 (20, 32) | <0.001 |
| Life expectancy, yrs | 81 (77, 83) | 72 (65, 76) | <0.001 |
| GDP* per capita, ×1000 $US | 36 (25, 46) | 8 (3, 16) | <0.001 |
| Human development index | 0.90 (0.85, 0.93) | 0.70 (0.55, 0.78) | <0.001 |
| Population density (people/km$^2$) | 108 (45, 214) | 77 (36, 209) | 0.201 |
| Mean stringency index | 53 (44, 58) | 54 (41, 64) | 0.495 |

*Gross domestic product.

countries, which mostly use OPV, compared to high-income nations, which mostly use IPV. Difference in the climate of countries using IPV *vs* those using OPV could be another cause of the observed difference [21].

To avoid rare cases of vaccine-associated paralytic polio (VAPP), most countries with higher HDI switched from OPV to IPV after they became free from wild poliovirus circulation. Most countries that are still using OPV are also free from the wild virus, but cannot switch to using IPV only because of the economic and logistic considerations. For this reason, IPV-only

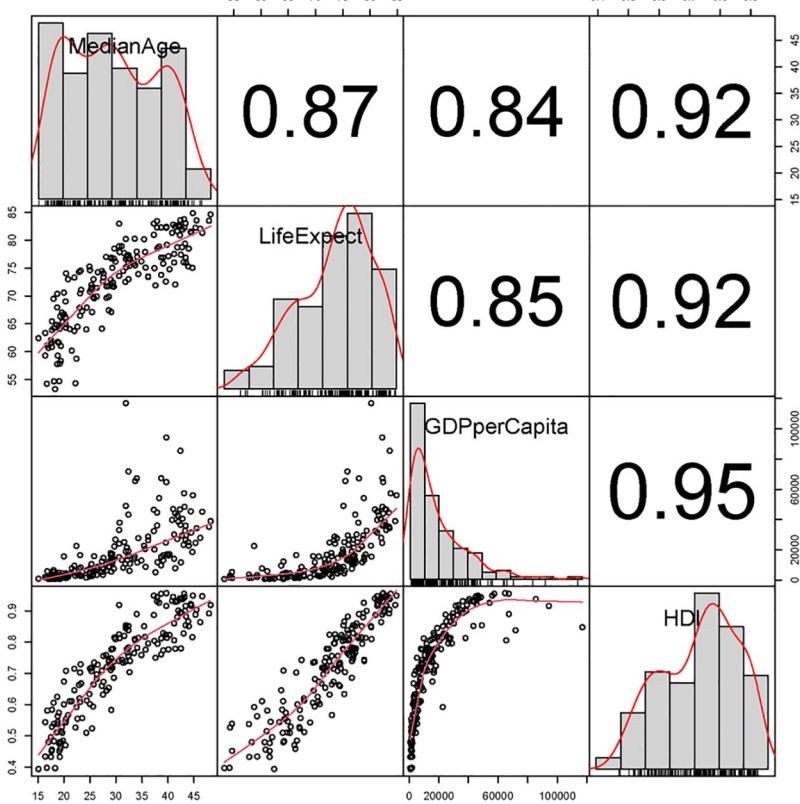

**Fig 3. Distribution of studied continuous variables and their correlation with each other.** Values are Spearman's $\rho$. The median age (MedianAge), life expectancy at birth (LifeExpect), and the gross domestic product per capita (GDPperCapita), all had a significant (p<0.001) high correlation with the human development index (HDI).

**Table 2. Results of negative binomial regression.**

| Variable | Coefficient (95% CI) | Adj IRR* (95% CI) | p value |
|---|---|---|---|
| HDI | 4.82 (0.18 to 9.46) | 123.37 (1.19 to 12765.75) | 0.042 |
| OPV vaccine[†] | -4.58 (-8.80 to -0.36) | 0.01 (0.00 to 0.70) | 0.033 |
| Interaction[‡] | 5.84 (0.98 to 10.69) | 342.15 (2.66 to 44008.53) | 0.019 |
| Intercept | 4.21 (0.10 to 8.32) | 67.17 (1.10 to 4086.18) | 0.045 |

*Adjusted incidence rate ratio.
[†]Using OPV vaccine compared to IPV (reference) (OPV = 1, IPV = 0).
[‡]HDI × Type of vaccine (OPV = 1, IPV = 0).
Nagelkerke's $R^2$ = 0.734.

countries have a higher GDP per capita and expectedly, should have a better health care infra-structure (as evident by higher life expectancy at birth and median age of their population, Table 1). A better infrastructure expectedly leads to a better and early diagnosis of the disease (more tests, having active surveillance, *etc.*) that results in a higher number of cases diagnosed, CP100K. At first glance, the under-diagnosis of cases with COVID-19 in low- and middle-income countries might explain the significantly lower observed CP100K in these countries where mostly use OPV (Figs 1 and 2). However, after taking into account the effect of HDI, a variable highly correlated with the level of investment in health care and the necessary infra-structure in a given country, use of OPV still remained an independent predictor of lower CP100K (Table 2).

The effects of lockdowns and international travel bans were hard to account for in our study since policies changed with time. However, the mean stringency index, an index reflect-ing the level of restrictions imposed in a given country, was not significantly different between countries using OPV *vs* those using IPV (Table 1). The population density was also not signifi-cantly different between the two groups of countries. None of these variables was thus taken into account in the model. HDI exhibits a strong correlation with GDP per capita, life expec-tancy at birth, and the median age of population in a country (Fig 3). This high correlation was not surprising because HDI, a measure reflecting the average wellbeing of people in a country, is in fact calculated based on the life expectancy at birth (and thus, the median age of the popu-lation), gross national income (and thus, the GDP) per capita, and other factors. The type of polio vaccine used in a given country strongly depends on the HDI—countries with higher HDI prefer and can afford to use IPV, which is significantly more expensive than OPV. This explains the significant interaction between HDI and the type of vaccine used observed in the regression model (Table 2). This might also explain the higher impact of each 0.1 unit increase in HDI on CP100K in countries using OPV (2.90-fold increase) compared to that in countries using IPV (1.62-fold increase); an 0.1 unit increase in HDI in low- and middle-income coun-tries (mostly using OPV), where the HDI is relatively low (median 0.7), would expectedly result in a more tangible improvement in the health infrastructure and reporting systems comapred to the same increase in HDI in high-income countries (mostly using IPV), where the HDI is relatively high (median 0.9) and a good health infrastructure has already been established.

The type of vaccine used is another independent variable found significantly associated with CP100K; countries using OPV had a lower CP100K compared to those using IPV (Table 2). We hypothesize that this correlation might reflect the protective effect of the live attenuated poliovirus in the OPV against SARS-CoV-2. Recently, we have shown that moth-ers whose children recently received OPV do not develop symptomatic COVID-19 for at

least 6 months, probably for the indirect exposure they had to the attenuated poliovirus in the vaccine [17].

Because of its nature, ecological studies cannot prove causal relationships; they are mostly considered "hypothesis generating" studies. In this case, the hypothesis about protective effect of OPV against COVID-19 is also supported by other independent evidence [17]. The non-specific protective effects provided by LAVs against unrelated infections have been demonstrated previously [8–11]. The mechanism involved is believed to be mediated by stimulation of the host innate immune system, including interferon production [5,7]. OPV is given mostly to infants who shed significant amounts of vaccine poliovirus in stool, and thus can transmit it to their caregivers and other close contacts. This can initiate chains of transmission resulting in exposure of a significant number of people to attenuated poliovirus, explaining why the virus can be readily found in the environment in countries using OPV [22]. This immunization through secondary exposure to vaccine virus creates strong herd immunity against poliovirus, and may also contribute to higher resistance to other infections by stimulating innate immunity. The community transmission of the virus is expected to be lower in countries with higher sanitation and better health care infrastructures, which are strongly correlated with higher HDI. This could support our hypothesis that OPV might provide protection against SARS-CoV-2 and be an additional reason why the protective effect of OPV was lower in countries with higher HDI (with presumably better sanitation) compared with communities where the virus can transmit more effectively.

An important but yet unknown aspect of the secondary effect of LAVs on susceptibility to unrelated infections is the duration of the protection. Interferon production is a transient phenomenon, but activation of other innate immunity pathways can be more durable. After immunization with OPV, the attenuated poliovirus can trigger epigenetic changes through methylation and acetylation of nuclear histones, marking the genes necessary for the host defense and making them more readily available to have stronger expression upon subsequent stimuli by viral particles, say SARS-CoV-2 [7,23]. This phenomenon, the so-called "trained immunity," could produce long-lasting effects reducing susceptibility to infections at the individual level; it can also reduce circulation of pathogens in the population.

A recently published study could not find any evidence for the protective effect of polio vaccination against SARS-CoV-2 infection [24]. However, authors of this study failed to distinguish between OPV (a live vaccine) and IPV (inactivated vaccine). Since non-LAVs were shown to lack the non-specific protective effects demonstrated for LAVs, no conclusion can be made from this study regarding use of an LAV like OPV.

An important consideration is whether the use of OPV against COVID-19 is a viable option, because many countries have stopped its use due to its ability to cause rare but serious VAPP cases and trigger the generation of circulating vaccine-derived polioviruses (cVDPV). Therefore, reintroduction of OPV into such countries should be considered very carefully. Furthermore, before this is done, direct clinical evidence of the effectiveness of OPV (and for this matter, other live vaccines) must be generated. This could be done in countries that still use OPV in their immunization programs. If successful, this could open several possibilities. Besides the classical OPV developed based on inherently genetically unstable Sabin strains, rationally designed novel OPV was recently created by targeted genetic manipulation to prevent its reversion to virulence [25]. In addition, a new class of broadly-specific vaccines employing innate immunity stimulation could be created for rapid response to emerging infections, before traditional pathogen-specific vaccines can be developed and introduced. They could become an important instrument in the pandemic preparedness toolbox.

## Limitations

The most important limitation of this study is that ecological studies demonstrating correlations cannot provide the final proof of causality, but rather should be used as hypothesis-generating tools. In this particular case, the correlation presented in this study agrees well with other observations of broadly protective effects of OPV against influenza [15,16] and other infections [17], making our conclusions more plausible. Another limitation of our study is that we did not consider the number of diagnostic tests daily performed in each country. That might be a better index compared to HDI for assessment of the health care quality in a given country. However, this information is missing for many countries and we thus decided to use HDI, which was available for most countries in the database we used. Furthermore, the model seems to be good; it could already explain more than 70% of the variance observed in CP100K. We also did not consider the coverage of vaccination against COVID-19 in our study. However, at the time of our data collection, most low- and middle-income countries either did not introduce COVID-19 vaccines or immunized a very small part of their populations. In contrast, high-income courtiers had a much more successful rollout of these vaccines, that are expected to reduce COVID-19 morbidity. Taking this factor into account would further strengthen our hypothesis that at a population level, use of OPV might reduce the SARS-CoV-2 incidence. Nor did we consider the coverage of other LAVs in the studied countries. It has been shown that many of LAVs such as MMR and BCG, can provide non-specific protection against other infections [5,10,26]. However, the coverage of these vaccines was not much different in the studied countries. BCG is routinely administered to all people in almost 90% of countries in the world; in some countries only at-risk groups are vaccinated [27]. On the other hand, while the attenuated poliovirus in the OPV can contaminate the close contacts of the vaccinated child (particularly where the hygienic standards are low) and thereby protect them from other infections [17], MMR and BCG can only affect the vaccinated person. Furthermore, recent experimental evidence suggests that antibodies generated in response to other viral pathogens may cross react with SARS-COV-2 and provide heterologous protection [28]. Controlling these covariates is complicated mainly because of the nature of our study—the inherent inability to make individual level causal inferences based on the aggregate data used in an ecological study.

## Conclusions

Use of OPV for routine immunization was associated with a lower incidence of COVID-19 in countries using OPV than in those using only IPV. We hypothesize that the live attenuated poliovirus in the OPV might protect people against SARS-CoV-2. Although this hypothesis has recently been supported by a cohort study [17], it should be tested in well-controlled clinical trials. Besides the possibility of using traditional or novel OPV to mitigate COVID-19, the results of such studies could open the possibility of creation of a new class of broadly specific vaccines, alongside the traditional antigen-specific vaccines.

## Supporting information

**S1 File.**
(CSV)

## Author Contributions

**Conceptualization:** Farrokh Habibzadeh, Konstantin Chumakov.

**Data curation:** Farrokh Habibzadeh, Konstantin Chumakov, Mahboobeh Yadollahie, Ashraf Simi.

**Formal analysis:** Farrokh Habibzadeh, Konstantin Chumakov, Mahboobeh Yadollahie, Kristen Stafford, Ashraf Simi, Iman Hafizi-Rastani.

**Investigation:** Farrokh Habibzadeh, Mahboobeh Yadollahie, Ashraf Simi.

**Methodology:** Farrokh Habibzadeh, Konstantin Chumakov, Mohammad M. Sajadi, Mahboobeh Yadollahie, Kristen Stafford, Ashraf Simi, Shyamasundaran Kottilil, Iman Hafizi-Rastani, Robert C. Gallo.

**Project administration:** Farrokh Habibzadeh.

**Software:** Farrokh Habibzadeh.

**Supervision:** Farrokh Habibzadeh, Konstantin Chumakov, Robert C. Gallo.

**Validation:** Farrokh Habibzadeh, Konstantin Chumakov, Mahboobeh Yadollahie, Kristen Stafford, Ashraf Simi, Robert C. Gallo.

**Visualization:** Farrokh Habibzadeh.

**Writing – original draft:** Farrokh Habibzadeh, Konstantin Chumakov, Mohammad M. Sajadi, Mahboobeh Yadollahie, Kristen Stafford, Ashraf Simi, Shyamasundaran Kottilil, Robert C. Gallo.

**Writing – review & editing:** Farrokh Habibzadeh, Konstantin Chumakov, Mohammad M. Sajadi, Mahboobeh Yadollahie, Kristen Stafford, Ashraf Simi, Shyamasundaran Kottilil, Iman Hafizi-Rastani, Robert C. Gallo.

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
