## [Decision Letter · Decision Letter 0]

1 Dec 2021

PONE-D-21-33498Use of Oral Polio Vaccine and the Incidence of COVID-19 in the WorldPLOS ONE

Dear Dr. Robert C. Gallo,

Thank you for submitting your manuscript to PLOS ONE. After careful consideration, we feel that it has merit but does not fully meet PLOS ONE’s publication criteria as it currently stands. Therefore, we invite you to submit a revised version of the manuscript that addresses the points raised during the review process.

We look forward to receiving your revised manuscript.

Kind regards,

Wen-Wei Sung, M.D., Ph.D.

Academic Editor

PLOS ONE

Journal Requirements:

Additional Editor Comments:

This observational study provides a possible relationship between oral Polio vaccine and low incidence of COVID-19 infection. As there is no solid molecular evidence to support the relationship, more discussion and a list of limitations might be necessary for this revision.

Reviewers' comments:

Reviewer's Responses to Questions

**Comments to the Author**

1. Is the manuscript technically sound, and do the data support the conclusions?

Reviewer #1: Yes

Reviewer #2: Partly

2. Has the statistical analysis been performed appropriately and rigorously? 

Reviewer #1: Yes

Reviewer #2: Yes

3. Have the authors made all data underlying the findings in their manuscript fully available?

Reviewer #1: Yes

Reviewer #2: Yes

4. Is the manuscript presented in an intelligible fashion and written in standard English?

Reviewer #1: Yes

Reviewer #2: Yes

5. Review Comments to the Author

Reviewer #1: The authors performed a population study utilizing publicly available data on the number of COVID cases per country and their usage of oral poliovirus vaccine vs inactivated poliovirus vaccine, as well as their human development index. The authors' data support their conclusion, but there are many variables that would go into the spread of the virus during the pandemic beyond HDI, GDP, and overall health care.

The authors should add in population density as another variable to assess.

The authors need to fix figure 3 which is cutoff.

The authors should include additional discussion on the potential roles that lockdowns and international travel could have played in the observed differences.

The authors should specifically state whether the outliers shown in figure 2 were included in the analyses or excluded. If they were excluded, they should provide the statistical criteria for the exclusion, and provide a secondary graph with them included.

Reviewer #2: The overall observation of this study might be true but the role of OPV is more hypothetical. What the authors failed to conclude with sufficient evidence is whether the observed protection for SARS-CoV-2 in countries with OPV is really due to OPV, BCG (widely used in LMICs) or many other rampant viral and bacterial infection with high frequency. Its well documented that non specific innate immunity provides protection to some degree from several other pathogens. Not only innate immunity, recent experimental evidences suggest that antibodies generated (humorl response) in response to other viral pathogens such as HIV and Flu cross reacts with SARS-COV-2 and might provide protection indirectly through antibody dependent virus clearance rather than direct neutralization. To conclude the specific role of OPV, a more direct comparison need to be done within the same environmental setting by comparing the number of COVID-19 infection among the population with OPV and without OPV vaccination. The circulating OPV strain among the population through improper hygiene in LMICs is not expected to provide significant dosing for indirect protection against SARS-CoV-2.

6. PLOS authors have the option to publish the peer review history of their article (what does this mean?). If published, this will include your full peer review and any attached files.

Reviewer #1: No

Reviewer #2: No

---

## [Author Response · Author response to Decision Letter 0]

8 Dec 2021

We have submitted the responses as a separate file, but here, please find a copy of our responses.

Thank you very much for your fruitful comments.

Journal Requirements:

Complied.

The data used in the study are available as supplementary material to this article.

Thank you very much for your comment. Figure 1, the map, is not copyrighted. It was developed based on the data obtained from a public domain (http://www.naturalearthdata.com/).

Additional Editor Comments:

This observational study provides a possible relationship between oral Polio vaccine and low incidence of COVID-19 infection. As there is no solid molecular evidence to support the relationship, more discussion and a list of limitations might be necessary for this revision.

Thank you very much for your comment. Please note that the real molecular data are rarely available for causal relationships in such diseases, including todays SARS-CoV-2 and CoVID-19. Some (like HIV) are an exception. However, if by this you mean a reasonable and known molecular mechanism of a likely explanation, it would certainly be by activating TLR and related receptors reacting to PAMP signals and thereby activating IFN-Type 1 genes, which in turn affect various immediate antiviral pathways. NK cells are also activated. However, we certainly agree we have not shown that here. We have already mentioned limitations of the study. In the revised version, we clearly show the Limitations section. We also added the probable protective effect of other live attenuated vaccines such as BCG and the probable role of cross-reacting antibodies on the results, as other limitations of the study.

Reviewers' comments:

Reviewer's Responses to Questions

Comments to the Author

1. Is the manuscript technically sound, and do the data support the conclusions?

Reviewer #1: Yes

Reviewer #2: Partly

2. Has the statistical analysis been performed appropriately and rigorously?

Reviewer #1: Yes

Reviewer #2: Yes

3. Have the authors made all data underlying the findings in their manuscript fully available?

Reviewer #1: Yes

Reviewer #2: Yes

4. Is the manuscript presented in an intelligible fashion and written in standard English?

Reviewer #1: Yes

Reviewer #2: Yes

5. Review Comments to the Author

Reviewer #1: The authors performed a population study utilizing publicly available data on the number of COVID cases per country and their usage of oral poliovirus vaccine vs inactivated poliovirus vaccine, as well as their human development index. The authors' data support their conclusion, but there are many variables that would go into the spread of the virus during the pandemic beyond HDI, GDP, and overall health care.

The authors should add in population density as another variable to assess.

Thank you very much for your comment. We added the population density to our study. It was not significantly different (p=0.201) in countries using OPV vs IPV (Table 1). Nevertheless, we’ve also added it to the model but it did not change anything significantly; it was not a significant predictor in the model. Therefore, we decided not to change the model.

The authors need to fix figure 3 which is cutoff.

We appreciate this suggestion and have complied by making this change to Figure 3.

The authors should include additional discussion on the potential roles that lockdowns and international travel could have played in the observed differences.

Thanks for raising this important point. The effects of lockdowns and international travel bans were hard to account for in our study since policies changed with time. However, we added the mean stringency index, an index reflecting the level of restrictions imposed in a given country, to our study. The mean was not significantly different between countries using OPV vs those using IPV. We have now added this in detail in the methodology section, Table 1, and Discussion.

The authors should specifically state whether the outliers shown in figure 2 were included in the analyses or excluded. If they were excluded, they should provide the statistical criteria for the exclusion, and provide a secondary graph with them included.

Thank you very much for making this suggestion. All the outliers were included in the analyses.

Reviewer #2: The overall observation of this study might be true but the role of OPV is more hypothetical. What the authors failed to conclude with sufficient evidence is whether the observed protection for SARS-CoV-2 in countries with OPV is really due to OPV, BCG (widely used in LMICs) or many other rampant viral and bacterial infection with high frequency. Its well documented that non specific innate immunity provides protection to some degree from several other pathogens. Not only innate immunity, recent experimental evidences suggest that antibodies generated (humorl response) in response to other viral pathogens such as HIV and Flu cross reacts with SARS-COV-2 and might provide protection indirectly through antibody dependent virus clearance rather than direct neutralization. To conclude the specific role of OPV, a more direct comparison need to be done within the same environmental setting by comparing the number of COVID-19 infection among the population with OPV and without OPV vaccination. The circulating OPV strain among the population through improper hygiene in LMICs is not expected to provide significant dosing for indirect protection against SARS-CoV-2.

Thank you very much. The reviewer has raised an important point. We agree this is in fact a limitation of this study, mainly due to the nature of the ecological studies. We addressed this point in the Limitations section of the Discussion.

---

## [Decision Letter · Decision Letter 1]

11 Feb 2022

PONE-D-21-33498R1Use of oral polio vaccine and the incidence of COVID-19 in the worldPLOS ONE

Dear Dr. Robert C. Gallo,

Thank you for submitting your manuscript to PLOS ONE. After careful consideration, we feel that it has merit but does not fully meet PLOS ONE’s publication criteria as it currently stands. Therefore, we invite you to submit a revised version of the manuscript that addresses the points raised during the review process.

In this revision, reviewer 2 suggests a second opinion cinsidering the revised MS was not fully responding to the comments. Therefore, I invited a new reviewer and further revision was needed.

We look forward to receiving your revised manuscript.

Kind regards,

Wen-Wei Sung, M.D., Ph.D.

Academic Editor

PLOS ONE

Reviewers' comments:

Reviewer's Responses to Questions

**Comments to the Author**

1. If the authors have adequately addressed your comments raised in a previous round of review and you feel that this manuscript is now acceptable for publication, you may indicate that here to bypass the “Comments to the Author” section, enter your conflict of interest statement in the “Confidential to Editor” section, and submit your "Accept" recommendation.

Reviewer #2: (No Response)

Reviewer #3: (No Response)

2. Is the manuscript technically sound, and do the data support the conclusions?

Reviewer #2: Partly

Reviewer #3: Partly

3. Has the statistical analysis been performed appropriately and rigorously? 

Reviewer #2: Yes

Reviewer #3: Yes

4. Have the authors made all data underlying the findings in their manuscript fully available?

Reviewer #2: Yes

Reviewer #3: Yes

5. Is the manuscript presented in an intelligible fashion and written in standard English?

Reviewer #2: Yes

Reviewer #3: Yes

6. Review Comments to the Author

Reviewer #2: (No Response)

Reviewer #3: This is a manuscript examining the correlation between the use of oral polio vaccine (OPV) and a reduced population case rate for COVID-19. As was noted in prior review, this study is correlative and hypothesis generating but does not provide a firm mechanism for the phenomenon other than “stimulation of the host innate immune system”. The analysis is simple and clear, demonstrating the correlation between OPV use and a lower incidence of diagnosed COVID-19 cases. As noted in the discussion, there are many potential explanations for these findings that are not directly associated with the action of OPV, and while not exhaustive, the points are made.

I have several concerns.

1. This isn’t a meta-analysis in the strict sense of being a comparison of point estimates in randomized controlled trials. This paper is an analysis of aggregated data that were scraped from publicly available databases. If this use of “meta-analysis” is acceptable to PLoS One, I defer to the editorial decision. That being said, there is no shame in analyzing data scraped from public sources—this is how weather forecasting is done—and it would be more appropriate to call this what it is: an analysis of publicly available data.

2. Figure 2, while simple, obscures the distribution of the data. It would be helpful to show the underlying datapoints (whether in the main figure or in a supplemental figure) to emphasize the clustering of datapoints.

More importantly, I am concerned about features of the analysis overall. Specifically, I am concerned that the conclusions drawn may be examples of Simpson’s paradox, specifically because A) there is a significant geographic split between the countries in the two groups and B) populations of radically different sizes are being compared as single datapoints. That being said, the analysis as presented is reasonable, but certainly subject to significant caveats.

3. In the second paragraph of the discussion, it is stated, “To avoid rare cases of vaccine-associate paralytic polio (VAPP), most countries with higher HDI prefer to use IPV instead of OPV. Countries using IPV are thus free from poliovirus circulation.” This is true but confused. It is true that the exclusive use of IPV prevents the circulation of vaccine type poliovirus (and rare revertant cases) in those countries, but the decision to make that change was based on the disappearance of wild-type poliovirus. For example, in the US the change was made after wild-type poliovirus circulation was eliminated in the Western hemisphere, thus tipping the risk-benefit ratio away from OPV use (see MMWR v46 RR-3 (1997)). The elimination of wild-type poliovirus circulation is certainly correlated with health care infrastructure, and it is highly likely that the continued use of OPV has more to do with vaccine distribution networks and the ease of OPV use in those settings.

4. The major problem with the conclusions of the paper is the fallacy that correlation equals causation. This study is a correlative, hypothesis generating study. It should not be sold as anything more than that. Generating hypotheses is important, but stating a correlation is not proof of mechanism.

5. The final conclusion, that vaccination with OPV could protect against SARS-CoV-2, is a vast overreach. That conclusion ignores the potential negative effects of unleashing poliovirus shedding in populations where vulnerable persons are at risk. There is an ever-growing population of people on immune modulating therapies, infants at risk from revertant poliovirus, and others who could have a negative outcome. The use of IPV in countries that can provide it was based on sound public health principles—suggesting a drastic change needs to have a similar level of care.

And perhaps more importantly, do the authors really think this is a viable solution? For example, in the US, poor vaccination rates have nothing to do with availability. I highly doubt that people who refuse a COVID vaccine will line up to take OPV, and they will be even more incensed at the idea of unleashing another virus over which they have no control.

Minor comments.

1. Page 6, this sentence is confusing / poorly worded / has an oddly placed comma.

The distribution of all studied continuous variables but the population density and the stringency index, was significantly different in countries using IPV compared with OPV (Table 1).

2. Page 6, “overdispersion” is misspelled.

7. PLOS authors have the option to publish the peer review history of their article (what does this mean?). If published, this will include your full peer review and any attached files.

Reviewer #2: No

Reviewer #3: No

---

## [Author Response · Author response to Decision Letter 1]

14 Feb 2022

Comments to the Author

We would like to take this opportunity to thank the respected reviewers for their fruitful comments. We revised the manuscript according to their comments. Now, we believe that the revised version has substantially been improved and is much better than before. 

6. Review Comments to the Author

Reviewer #2: (No Response)

Reviewer #3: This is a manuscript examining the correlation between the use of oral polio vaccine (OPV) and a reduced population case rate for COVID-19. As was noted in prior review, this study is correlative and hypothesis generating but does not provide a firm mechanism for the phenomenon other than “stimulation of the host innate immune system”. The analysis is simple and clear, demonstrating the correlation between OPV use and a lower incidence of diagnosed COVID-19 cases. As noted in the discussion, there are many potential explanations for these findings that are not directly associated with the action of OPV, and while not exhaustive, the points are made.

I have several concerns.

1. This isn’t a meta-analysis in the strict sense of being a comparison of point estimates in randomized controlled trials. This paper is an analysis of aggregated data that were scraped from publicly available databases. If this use of “meta-analysis” is acceptable to PLoS One, I defer to the editorial decision. That being said, there is no shame in analyzing data scraped from public sources—this is how weather forecasting is done—and it would be more appropriate to call this what it is: an analysis of publicly available data.

Thank you very much for pointing this out. We do completely agree with you that the term “meta-analysis” in the sense that you mentioned above should not be used in this context. We prefer to use the term “systematic review with meta-analysis” for the purpose the respected referee mentioned. The term meta-analysis was used in this manuscript to describe the set of statistical methods used to aggregate data obtained from various sources. We changed the statements to reflect this point and hope that, if necessary, journal editors could perform further editing according to the journal’s style. 

2. Figure 2, while simple, obscures the distribution of the data. It would be helpful to show the underlying datapoints (whether in the main figure or in a supplemental figure) to emphasize the clustering of datapoints.

Thank you very much for raising this important point. Data points were added to the main figure, as suggested. 

More importantly, I am concerned about features of the analysis overall. Specifically, I am concerned that the conclusions drawn may be examples of Simpson’s paradox, specifically because A) there is a significant geographic split between the countries in the two groups and B) populations of radically different sizes are being compared as single datapoints. That being said, the analysis as presented is reasonable, but certainly subject to significant caveats.

Thank you very much for the important point you raised. Based on the distributions of the data points in countries using IPV and OPV (see revised Fig 2) and taking into account that we tried to identify and control, as much as possible, the confounding factors and covariates in the final model, it is unlikely that the observed findings resulted from Simpson’s paradox. You are completely correct that the population sizes were significantly different, but we tried to address this disparity by comparing the cumulative incidence of the disease (corrected for the population size) rather than the number of infections. We also tried to adjust the model for the marked differences observed between the countries in the two groups by considering a number of covariates the most important of which was the Human Development Index (HDI). Anyway, Figure 2 presents the results of a univariate analysis and we did not rely on it. We mainly used the results obtained from the regression model. Nevertheless, we have strengthened in the Discussion the point about possible caveats.

3. In the second paragraph of the discussion, it is stated, “To avoid rare cases of vaccine-associate paralytic polio (VAPP), most countries with higher HDI prefer to use IPV instead of OPV. Countries using IPV are thus free from poliovirus circulation.” This is true but confused. It is true that the exclusive use of IPV prevents the circulation of vaccine type poliovirus (and rare revertant cases) in those countries, but the decision to make that change was based on the disappearance of wild-type poliovirus. For example, in the US the change was made after wild-type poliovirus circulation was eliminated in the Western hemisphere, thus tipping the risk-benefit ratio away from OPV use (see MMWR v46 RR-3 (1997)). The elimination of wild-type poliovirus circulation is certainly correlated with health care infrastructure, and it is highly likely that the continued use of OPV has more to do with vaccine distribution networks and the ease of OPV use in those settings.

We appreciate this comment and agree that the sentence is confusing. While the main reason for the switch from OPV to IPV was triggered by the need to stop VAPP, it was predicated on the absence of wild poliovirus circulation. The situation in less affluent countries is slightly different. While in most countries where OPV is still being used wild-type poliovirus circulation was stopped many years ago, and VAPP and cVDPV present a problem, the continued use of OPV in these countries is related to economic and logistical considerations. We have edited this sentence to make it less confusing.

4. The major problem with the conclusions of the paper is the fallacy that correlation equals causation. This study is a correlative, hypothesis generating study. It should not be sold as anything more than that. Generating hypotheses is important, but stating a correlation is not proof of mechanism.

Thank you very much for raising this very important point. We do agree with you that an ecological study cannot provide any cause-and-effect relationship. It can just provide association. We have revised different parts of the manuscript, played down any causal inference mentioned and exclusively stated that that’s all a hypothesis that might be correct and although a recent cohort study has supported this hypothesis, it should be tested in a clinical trial.

5. The final conclusion, that vaccination with OPV could protect against SARS-CoV-2, is a vast overreach. That conclusion ignores the potential negative effects of unleashing poliovirus shedding in populations where vulnerable persons are at risk. There is an ever-growing population of people on immune modulating therapies, infants at risk from revertant poliovirus, and others who could have a negative outcome. The use of IPV in countries that can provide it was based on sound public health principles—suggesting a drastic change needs to have a similar level of care.

Thank you very much for your comment, this is indeed an important point. The risk of OPV-associated complications must be considered before the decisions about the mass use of OPV is made. In this paper we do not propose this but rather call for prospective clinical trials to prove the beneficial effect of OPV use against COVID-19 and potentially other emerging infections. If it is confirmed, other possibilities are available, including the use if safer novel OPV that was recently developed. We have modified this Discussion to reflect this point. 

And perhaps more importantly, do the authors really think this is a viable solution? For example, in the US, poor vaccination rates have nothing to do with availability. I highly doubt that people who refuse a COVID vaccine will line up to take OPV, and they will be even more incensed at the idea of unleashing another virus over which they have no control.

We would like to thank the reviewer for this comment, and regret that we came across in a simplistic way as if we propose immediate use of OPV as a vaccine against COVID-19. The main point that we want to make is that broadly-specific protective effects of live vaccines are totally underappreciated and deserve to be closely studied in well-controlled prospective clinical trials. While the wisdom of reintroduction of OPV in the US and most European countries may be questioned, there are countries where such trials will not pose a great risk, if conducted properly. The information generated in such studies could be extremely valuable and be used for creation of a new kind of broadly specific vaccines, alongside the traditional pathogen-specific ones. We have added the discussion of this point in the Discussion, to better reflect our position. 

Minor comments.

1. Page 6, this sentence is confusing / poorly worded / has an oddly placed comma.

The distribution of all studied continuous variables but the population density and the stringency index, was significantly different in countries using IPV compared with OPV (Table 1).

Thank you very much for pointing out this ambiguous statement. It was reworded as:

The distributions of the median age, life expectancy, and GDP per capita were significantly different between countries using IPV and OPV (Table 1). The median HDI in countries using IPV was significantly (0.90 vs 0.70, p<0.001) higher than that in countries using OPV. The population density and the mean stringency index were not significantly different between countries using OPV and those using IPV.

2. Page 6, “overdispersion” is misspelled.

Thank you very much for pointing out that mistake. It was corrected.

7. PLOS authors have the option to publish the peer review history of their article (what does this mean?). If published, this will include your full peer review and any attached files.

Do you want your identity to be public for this peer review? For information about this choice, including consent withdrawal, please see our Privacy Policy.

Reviewer #2: No

Reviewer #3: No

---

## [Decision Letter · Decision Letter 2]

4 Mar 2022

Use of oral polio vaccine and the incidence of COVID-19 in the world

PONE-D-21-33498R2

Dear Dr. Robert C. Gallo,

We’re pleased to inform you that your manuscript has been judged scientifically suitable for publication and will be formally accepted for publication once it meets all outstanding technical requirements.

Kind regards,

Wen-Wei Sung, M.D., Ph.D.

Academic Editor

PLOS ONE

Reviewers' comments:

Reviewer's Responses to Questions

**Comments to the Author**

1. If the authors have adequately addressed your comments raised in a previous round of review and you feel that this manuscript is now acceptable for publication, you may indicate that here to bypass the “Comments to the Author” section, enter your conflict of interest statement in the “Confidential to Editor” section, and submit your "Accept" recommendation.

Reviewer #3: All comments have been addressed

2. Is the manuscript technically sound, and do the data support the conclusions?

Reviewer #3: Yes

3. Has the statistical analysis been performed appropriately and rigorously? 

Reviewer #3: Yes

4. Have the authors made all data underlying the findings in their manuscript fully available?

Reviewer #3: Yes

5. Is the manuscript presented in an intelligible fashion and written in standard English?

Reviewer #3: Yes

6. Review Comments to the Author

Reviewer #3: I applaud the authors for addressing all of my concerns, including my strident ones. I appreciate their work.

7. PLOS authors have the option to publish the peer review history of their article (what does this mean?). If published, this will include your full peer review and any attached files.

Reviewer #3: **Yes: **M. Anthony Moody

---

## [Editor Report · Acceptance letter]

9 Mar 2022

PONE-D-21-33498R2 

Use of oral polio vaccine and the incidence of COVID-19 in the world 

Dear Dr. Gallo:

I'm pleased to inform you that your manuscript has been deemed suitable for publication in PLOS ONE. Congratulations! Your manuscript is now with our production department. 

Kind regards, 

on behalf of

Dr. Wen-Wei Sung 

Academic Editor

PLOS ONE